# DiffHarmony++: Enhancing Image Harmonization with Harmony-VAE and Inverse Harmonization Model

## ABSTRACT

Latent diffusion model has demonstrated impressive efficacy in image generation and editing tasks. Recently, it has also promoted the advancement of image harmonization. However, methods involving latent diffusion model all face a common challenge: the severe image distortion introduced by the VAE component, while image harmonization is a low-level image processing task that relies on pixel-level evaluation metrics. In this paper, we propose Harmony-VAE, leveraging the input of the harmonization task itself to enhance the quality of decoded images. The input involving composite image contains the precise pixel level information, which can complement the correct foreground appearance and color information contained in denoised latents. Meanwhile, the inherent generative nature of diffusion models makes it naturally adapt to inverse image harmonization, i.e. generating synthetic composite images based on real images and foreground masks. We train an inverse harmonization diffusion model to perform data augmentation on two subsets of iHarmony4 and construct a new human harmonization dataset with prominent foreground objects. Extensive experiments demonstrate the effectiveness of our proposed Harmony-VAE and inverse harmonization model. The code, pretrained models and the new dataset will be made publicly available.

## CCS CONCEPTS

• **Computing methodologies → Computer vision tasks**.

## KEYWORDS

image harmonization, latent diffusion model, VAE, data augmentation, inverse harmonization, stable diffusion

## 1 INTRODUCTION

Image composition is a crucial technique in digital editing, where a new image is created by combining the foreground of one image with the background of another. This process, widely applied in fields such as advertising and entertainment, often encounters a challenge: the merged image may appear unrealistic due to mismatches in lighting and color between the foreground and background. To solve this problem, image harmonization is employed. It involves modifying the foreground to ensure visual consistency with the background, focusing on aligning color, illumination, and texture, without changing the original content or meaning. Recent

**Unpublished working draft. Not for distribution.**

advancements in deep learning [45, 11, 17, 41] have significantly facilitated image harmonization.

The latent diffusion model (LDM) [37] is gaining increasing attention and has established new SoTA in the field of image generation. It can swiftly transfer to downstream image-to-image translation tasks such as image editing [1, 23] and restoration [33, 47] . As image harmonization can be categorized as an image-to-image translation task, the latent diffusion model is also suitable for doing image harmonization. For example, Li et al. [27] follow the network architecture in ControlNet [51] and adapt pretrained latent diffusion model, i.e. Stable Diffusion, to perform image harmonization, but they ignore the image distortion problem inherent in VAE decoding process, resulting in unsatisfactory model performance. Zhou et al. [52] adopt a two-stage approach. The first stage finetunes the denoising UNet, while the second stage involves designing a refinement module to alleviate the image distortion issue. It finally achieves SoTA performance. However, this approach exhibits several limitations: 1) The second-stage training relies on the results of the first stage, necessitating retraining of the second stage upon updates in the first stage. 2) Introducing an independent UNet model complicates the harmonization process. 3) It's limited to usage at fixed resolutions.

To solve aforementioned limitations, we propose Harmony-VAE, which incorporates composite images as additional information into the decoding process of the VAE, enabling the VAE decoder to reconstruct images with realistic details. Our motivation stems from the highly precise object shape information present in composite images, which can complement the correct foreground appearance and color information contained in denoised latents. The training of Harmony-VAE is independent of fine-tuning denoising UNet; it reconstructs real images with composite images as condition, significantly reducing training costs, and it can improve the performance of all LDM-based harmonization models. Furthermore, although trained only on 256px images, Harmony-VAE significantly enhances the decoding quality of higher resolution images, demonstrating strong generalization capabilities. We call DiffHarmony equipped with Harmony-VAE as DiffHarmony++.

The latent diffusion model enhanced by our proposed Harmony-VAE, with simple modifications, can be adapted to construct a model that performs inverse image harmonization, serving the purpose of data augmentation. Inverse image harmonization refers to the generation of synthetic composite images from real images and foreground masks. The outcomes should not be unique, as the content of the images may be influenced by various factors such as weather and lighting conditions. This process can be modeled as a one-to-many image-to-image translation task, and employing a diffusion model to handle it is a very natural choice. Based on DiffHarmony++, we train an inverse harmonization diffusion model on iHarmony4[10]. This model enables virtually unlimited expansion of harmonization data. Leveraging the inverse harmonization

model, we perform data augmentation on two smaller subsets of iHarmony4, namely Hday2night and HFlickr. Compared to training solely on the original data, training with augmented data greatly enhances performance on these two datasets.

Another significant advantage of our proposed inverse harmonization model is to automatically construct new image harmonization datasets, which has long been a challenge in the field of image harmonization. The creation of datasets like RealHM[20] often needs extensive manual effort and expertise in digital image processing, proving to be time-consuming and labor-intensive. iHarmony4 employs automatic color transfer algorithms to generate synthetic composite images, but still need manual screening to filter out unreasonable data. Leveraging the inverse harmonization model, we generate a substantial number of synthetic composite images based on the imaterialist[12] dataset and train a harmony classifier to identify most unharmonized images, culminating in the creation of the Human Harmony dataset. Training harmonization models on this newly constructed dataset, both qualitative and quantitative evaluations affirm that the proposed inverse harmonization diffusion model is a highly promising approach for building new datasets.

Our contributions can be summarized as follows:

- We propose Harmony-VAE to tackle the image distortion problem inherent in VAE decoding process, which is the most challenging aspect of applying LDM to image harmonization tasks.
- We design a simple yet effective inverse harmonization diffusion model, and validate its efficacy on two image harmonization datasets.
- We contribute a new Human Harmony dataset with inverse harmonization model and use harmony classifier to further filter out high quality data.

## 2 RELATED WORKS

### 2.1 Image Harmonization

Image harmonization is a critical task in image composition [34], aimed at achieving visual consistency between the foreground and background of composite images. Early efforts were primarily centered around traditional color matching algorithms [49, 43, 44, 50, 7]. These methods focused on aligning the low-level color statistics between the foreground and background, utilizing techniques such as shifting, scaling, and histogram matching. For example, Lalonde et al. [26] recolor image regions for realistic compositing by studying the color statistics of a large dataset of natural images and looking at differences in color distribution in realistic and unrealistic images. With the advent of deep learning, a surge of supervised deep harmonization methods came out [45, 11, 17, 41, 36, 46, 48, 53, 20, 2, 3, 4]. Tsai et al. [45] propose an end-to-end deep convolutional neural network for image harmonization, which can capture both the context and semantic information of the composite images during harmonization. Cun et al. [11] learn the feature map in the masked region and the others individually with a novel attention module named Spatial-Separated Attention Module. The application of domain translation or style transfer techniques to image harmonization [8, 16, 30, 10] offers a creative means of reconciling

discrepancies between the foreground and background by conceptualizing different illumination conditions as distinct domains or styles. Cong et al. [10] translate foreground to the same domain as background with a domain verification discriminator. Cong et al. [8] use a domain code extractor to capture the background domain information to guide the foreground harmonization, which is regulated by well-tailored triplet losses. In the mean time, the integration of Retinex theory [15, 14, 13] into image harmonization has opened new pathways by decomposing images into reflectance and illumination maps, and also the development of deep networks predicting color transformation [9, 22, 29, 49] signifies a balance between efficiency and effectiveness, streamlining the harmonization process without sacrificing quality.

### 2.2 Image Harmonization Dataset

Jiang et al. [20] construct RealHM dataset with 216 image pairs by manually adjusting the foreground according to the background, which is time-consuming, labor-intensive, and unreliable. Another line of work is collecting a set of foreground images captured in different illumination conditions, followed by replacing one foreground with another counterpart, for example Transient Attributes Database [25] (101 sets, in which each set has well-aligned images for the same scene captured in different conditions). Cao et al. [3] and Guo et al. [15] try to construct harmonization dataset by varying the lighting condition of the same scene using 3D rendering techniques, however the rendered images have large domain gap with real images so the trained model is hard to directly applied to real test images. Some other works [45, 11, 10] adopted an inverse approach, i.e., adjusting the foreground of real image to create synthetic composite image. The representative, iHarmony4 dataset utilize automatic color transfer for foreground adjustment followed by aesthetic predictor and binary CNN classifier for filtering (also at last need manual screening). Niu et al. [35] learns a VAE conditioned on ground-truth real image and foreground mask to predict color transformation LUT to adjust foreground part in real image, producing new synthetic composite images automatically.

### 2.3 Diffusion Model

Diffusion models are adept at generating realistic images from random noise, showcasing unparalleled performance in image synthesis. Ho et al. [19] introduced Denoising Diffusion Probabilistic Models (DDPMs) employing a Markovian diffusion process. This method progressively adds noise to an image until it becomes pure noise. Then, a deep neural network trained to predict this noise reverses the process, creating a new image from the noise. Alternatively, a non-Markovian diffusion process [42] is proposed to offer a quicker and more adaptable solution than the Markovian method in DDPMs. Due to the remarkable ability to create lifelike images, diffusion models have been extensively used in various image synthesis applications by researchers. For example, Palette [40], a conditional diffusion model set new standards in image-to-image translation tasks including colorization, inpainting, uncropping, and JPEG restoration. RePaint [32] utilized a pre-trained unconditional DDPM as a basis for generation, conditioning the process by sampling from unmasked regions of the provided image data.

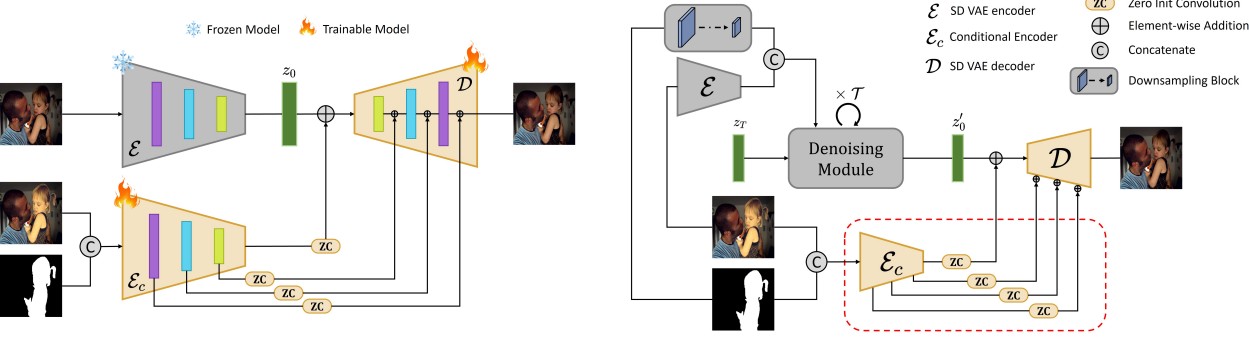

**(a) Training of Harmony-VAE**    **(b) DiffHarmony++ Inference**

**Figure 1: Demonstration of Harmony-VAE training and inference of DiffHarmony++. (a) The training goal of Harmony-VAE is to reconstructing $\hat{I}$ with $I$ and $M$ as condition. The extracted features will be fed into VAE decoder through skip connection in way of element-wise addition. We add zero-initialized convolution layer before each skip connection to stablize training. (b) At denoising stage, the process starts with pure Gaussian noise $z_T$, iteratively refines it through Denoising U-Net, and finally get denoised latent variable $z'_0$, during which the encoded composite image $\mathcal{E}(I)$ and downsampled foreground mask image are concatenated in channel dimension as condition. Then the latent variable $z'_0$ will be decoded into harmonized image $\tilde{I}$.**

Sahak et al. [39] introduced SR3+, a diffusion model that set new benchmarks in the blind super-resolution task.

Diffusion models in latent space have recently garnered unprecedented attention. Latent Diffusion Model (LDM) [37] is proposed to serve as the theoretical foundation for Stable Diffusion. Numerous works based on Stable Diffusion targeted on solving image-to-image translation tasks, and some have applied latent diffusion model framework to the image harmonization task (which can be categorized as image-to-image translation). For instance, Appearance Consistency Discriminator [27] is proposed to guide LDM and convert the generated image from RGB to HSV/HSL space to adjust the lightness channel. Chen et al. [6] achieved Zero-Shot Image Harmonization by incorporating Attention-Constraint Text and Content Retention modules into Stable Diffusion. Lu et al. [31] focused on using pretrained LDM to blend photographic objects into paintings, achieving artistically coherent composite images. Zhou et al. [52] proposed DiffHarmony, which adapted the Stable Diffusion inpainting variation for image harmonization and propose a refinement stage to address image distortion issues.

## 3 HARMONY-VAE

### 3.1 Preliminary

The Stable Diffusion (SD) model stands as a quintessential representation of latent diffusion model and serves as the cornerstone for our approach. It undergoes pre-training in two distinct stages employing a VAE (Variational AutoEncoder) and a denoising U-Net. In the first stage, it trains the VAE, where the encoder $\mathcal{E}$ first encodes the image $I$ into a latent space, resulting the latent variable $z_0 = \mathcal{E}(I)$; subsequently the decoder $\mathcal{D}$ endeavors to reconstruct it into the original image, yielding the reconstructed image $\hat{I}$. In the second stage, the VAE is frozen and the goal is training a denoising U-Net $\epsilon_\theta$, which involves adding noise over $t$ steps to the latent variable $z_0$ to get noisy latent $z_t$ ($t \in [1, T]$), and updating the denoising U-Net with latent denoising loss, as formulated below:

$$\mathcal{L} = \mathbb{E}_{z_0,c,\epsilon \sim \mathcal{N}(0,1),t} \left[ \|\epsilon - \epsilon_\theta(z_t, t, c)\|_2^2 \right] \quad (1)$$

Here, $\epsilon$ denotes the noise added to the latent variable $z_0$ at each noise step, $\epsilon_\theta$ represents the denoising U-Net, receiving timestep $t$ and additional conditions $c$ (e.g., text, conditional images, masks, etc.) as input.

During the inference process, pure noise $z_T$ is sampled from a normal distribution. The model will iteratively employ $\epsilon_\theta$ to estimate the noise for each denoising step $t$, progressively refining the latent variable $z_T$ to ultimately attain denoised latent variable $z'_0$. Finally, the denoised latent variable $z'_0$ is fed into the decoder $\mathcal{D}$ to generate the image. For intricate details on the training and inference of Stable Diffusion, please refer to [37].

In image harmonization task, we delineate the conceivable variables as follows: the input to image harmonization task comprises the composite image $I$ and the foreground mask image $M$ (where the foreground is represented as 1 in white, and the background as 0 in black), while the output is the harmonized image $\tilde{I}$. Ground-truth real image utilized in model training and evaluation is denoted as $\hat{I}$. The composite image $I$ can be divided into the foreground component $I_f$ and the background component $I_b$. The objective of image harmonization task is to adjust the composite foreground $I_f$ to produce the harmonized image $\tilde{I}$, which should closely resemble $\hat{I}$.

When employing the latent diffusion model for image harmonization, a viable approach could be: during training, setting $z_0 = \mathcal{E}(\hat{I})$, $c = (\mathcal{E}(I), \text{down}(M))$ (where down denotes the downsampling operation), in detailed words utilizing the encoded composite image and downsampled mask image as conditions (typically concatenated along the channel dimension) for denoising the noisy latent $z_t$. During inference, initialize from random noise $z_T$, progressively denoise it to obtain $z'_0$ and decode it through $\mathcal{D}$ to yield $\tilde{I}$. This approach has already been proven effective in DiffHarmony[52], and

in our subsequent descriptions, we assume that the latent diffusion model part adopts the same architecture as it.

## 3.2 Our Method

When employing latent diffusion models for image harmonization task, a common obstacle to model performance is the image distortion problem caused by VAE[24] decoding process. Image harmonization falls within the domain of low-level image processing, demanding pixel-level accuracy of output images. However, VAE in Stable Diffusion, which decodes a single latent variable to an image, often results in distortion in subtle high-frequency textures or lead to the fabrication of content not present in the original image (as VAE fundamentally operates as a generative model).

To address the aforementioned issue, we propose the Harmony-VAE, aiming at enhancing the quality of the output harmonized image $\tilde{I}$ during the decoding stage of VAE by utilizing the input of the harmonization task, namely the composite image $I$ and foreground mask image $M$.

Specifically, we incorporate an additional encoder, the conditional encoder $\mathcal{E}_c$, to encode $I$ and $M$. During the encoding process, we obtain the final latent variables and intermediate features of the encoder. These features are fused into VAE decoder through skip connections (similar to UNet[38]) and element-wise addition. The training objective of the Harmony-VAE is to reconstruct the real image $\hat{I}$ with $I$ and $M$ as condition. This process can be mathematically formalized as:

$$\mathcal{L} = \left\| \hat{I} - \mathcal{D}\left(\mathcal{E}(\hat{I}), \mathcal{E}_c(I, M)\right) \right\|_2^2 \tag{2}$$

Strictly speaking, during the training of the Harmony-VAE, $\mathcal{E}(\hat{I})$ part should actually be the latent variable $z'_0$. However, when the denoising module performs sufficiently well, the distributions of the latent variables corresponding to the real image $\hat{I}$ and the harmonized image $\tilde{I}$ are close enough, thereby using the encoded real image $\mathcal{E}(\hat{I})$ as the model input for training does not incur significant performance loss. Moreover, this choice brings another benefit: the training of the Harmony-VAE model does not depend on the denoising part, greatly reducing the training cost, as there is no need to generate data samples from the diffusion model offline or online.

During training of the Harmony-VAE, the weights of the conditional encoder $\mathcal{E}_c$ are initialized from the weights of the original VAE encoder. To maintain training stability, we add zero-initialized convolution layers before each skip connection, and utilize the zero-initialization strategy for the portion of convolution weights processing the mask image $M$ at the input convolution conv_in. During training, we set the weights of the conditional encoder $\mathcal{E}_c$, decoder $\mathcal{D}$, and all zero-initialized convolution layers trainable. After training, it can be integrated seamlessly into DiffHarmony inference process. We call DiffHarmony equipped with Harmony-VAE as DiffHarmony++. For a comprehensive schematic illustration of Harmony-VAE training and DiffHarmony++ inference, please refer to Figure 1. We highlight the newly added model weights in red dashed lines.

# 4 INVERSE IMAGE HARMONIZATION

## 4.1 Inverse Harmonization Model

The acquisition of training data for image harmonization task has always been a costly endeavor. Traditional procedures for constructing harmonization datasets often involve three main steps: selecting challenging compositing pairs, creating precise masks for foreground objects with distinct boundaries, and adjusting foreground appearance to match the background using tools like Photoshop [20]. These steps require a considerable amount of manual labor and expertise in image processing, making them challenging and time-consuming. The construction of the iHarmony4 [10] dataset employs a highly automated process, such as color transfer modules and classifiers to identify unreasonable data, but still requires manual screening.

Diffusion models are initially applied to image generation and conditional image generation. Considering harmonization data generation (as in iHarmony4), given the foreground mask image $M$ and the real image $\hat{I}$, the output is the composite image $I$. The resulting composite image $I$ is not unique because many variations can make the foreground appear incongruous with the background. This process can be modeled as a typical one-to-many conditional image-to-image translation task, hence employing diffusion models to generate harmonization data is a natural choice.

Our approach is based on DiffHarmony++ to train an inverse harmonization model. The model takes the foreground mask image $M$ and the real image $\hat{I}$ as input and outputs the composite image $I$. We train it using the existing harmonization dataset iHarmony4.

We can use the inverse harmonization model to generate additional training data for existing harmonization datasets. Specifically, we generate $K$ candidate composite images for each $(\hat{I}, M)$ data pair in the dataset and specify the blending ratio as $\gamma$. Assuming the number of $(\hat{I}, M)$ data pairs in the original dataset is $N_1$, and the total number of original composite images is $N_2$, we randomly select $\left\lceil \frac{\gamma N_2}{N_1} \right\rceil$ newly generated composite images for each $(\hat{I}, M)$ data pair and add them to the training data. After blending, we train new harmonization models separately using the original dataset and the augmented dataset, comparing the results of the two experiments on oiriginal test set to observe the benefits of data augmentation (Experimental results show that our data augmentation strategy significantly improves model performance compared to training solely on the original data. Details can be found in the 5.3 section).

## 4.2 Construct Human Harmony Dataset

The field of image harmonization has long lacked datasets specifically tailored to human portraits domains. These datasets have the potential to optimize product showcases and enhance user experiences, offering significant value and innovation opportunities for e-commerce, retail, and fashion design. However, for the reasons mentioned earlier, constructing such datasets has been challenging. Now, after verifying the effectiveness of our inverse harmonization model, we have the opportunity to construct a Human Harmony dataset in a cheap and fast way.

Furthermore, the analysis in DiffHarmony [52] suggests that latent diffusion models may perform better on samples with large foregrounds. Portrait photography data typically features large

| Dataset | Metric | Composite | DoveNet[10] | BargainNet[8] | RainNet[30] | iS²AM[41] | D-HT[14] | SCS-Co[16] | HDNet[5] | Li et al. [28] | GIFT[35] | DiffHarmony[52] | Ours |
|---|---|---|---|---|---|---|---|---|---|---|---|---|---|
| HCOCO | PSNR↑ | 35.47 | 35.83 | 37.03 | 37.08 | 39.16 | 38.76 | 39.88 | 41.04 | 34.33 | 39.91 | 41.71 | 42.42 |
| | MSE↓ | 41.07 | 36.72 | 24.84 | 29.52 | 16.48 | 16.89 | 13.58 | 11.60 | 59.55 | 12.70 | 9.18 | 8.43 |
| | fMSE↓ | 542.06 | 551.01 | 397.85 | 501.17 | 266.19 | 299.30 | 245.54 | - | - | 229.68 | 170.44 | 155.73 |
| HAdobe5k | PSNR↑ | 33.77 | 34.34 | 35.34 | 36.22 | 38.08 | 36.88 | 38.29 | 41.17 | 33.18 | 41.08 | 41.78 |
| | MSE↓ | 63.40 | 52.32 | 39.94 | 43.35 | 21.88 | 38.53 | 21.01 | 13.58 | 161.36 | 18.35 | 19.51 | 18.81 |
| | fMSE↓ | 404.62 | 380.39 | 279.66 | 317.55 | 173.96 | 265.11 | 165.48 | - | - | 143.96 | 120.78 | 113.18 |
| HFlickr | PSNR↑ | 30.03 | 30.21 | 31.34 | 31.64 | 33.56 | 33.13 | 34.22 | 35.81 | 29.21 | 34.44 | 37.10 | 37.74 |
| | MSE↓ | 143.45 | 133.14 | 97.32 | 110.59 | 69.97 | 74.51 | 55.83 | 47.39 | 224.05 | 54.33 | 30.89 | 28.77 |
| | fMSE↓ | 785.65 | 827.03 | 698.40 | 688.40 | 443.65 | 515.45 | 393.72 | - | - | 360.08 | 216.27 | 201.52 |
| Hday2night | PSNR↑ | 34.50 | 35.27 | 35.67 | 34.83 | 37.72 | 37.10 | 37.83 | 38.85 | 34.08 | 38.28 | 39.45 | 39.49 |
| | MSE↓ | 76.61 | 51.95 | 50.98 | 57.40 | 40.59 | 53.01 | 41.75 | 31.97 | 122.41 | 37.81 | 22.42 | 22.48 |
| | fMSE↓ | 989.07 | 1075.71 | 835.63 | 916.48 | 590.97 | 704.42 | 606.80 | - | - | 566.47 | 470.846 | 464.35 |
| Average | PSNR↑ | 34.35 | 34.76 | 35.88 | 36.12 | 38.19 | 37.55 | 38.75 | 40.46 | 32.70 | 38.92 | 40.97 | 41.66 |
| | MSE↓ | 59.67 | 52.33 | 37.82 | 40.29 | 24.44 | 30.30 | 21.33 | 16.55 | 141.84 | 19.46 | 14.86 | 13.98 |
| | fMSE↓ | 594.67 | 532.62 | 405.23 | 469.60 | 264.96 | 320.78 | 248.86 | - | - | 225.30 | 166.48 | 153.98 |

Table 1: Quantitative comparison across four sub-datasets of iHarmony4 and in general. Top two performance are shown in red and blue. ↑ means the higher the better, and ↓ means the lower the better.

foregrounds objects. Building a Human Harmony dataset will help us further validate this hypothesis.

We select imaterialist-fashion-2020-fgvc7 [12] as our initialization. It comprises a vast number of high-resolution portrait photographs, and the majority features large foregrounds objects (human body parts). Each image is paired with highly-detailed segmentation map, which can be used to construct accurate foreground mask.

To ensure the quality of the generated harmonization data, we train a harmony classifier using the iHarmony4 dataset. Given an input image, the classifier outputs the probability of it belonging to composite images.

Following Section 4.1, we generate $K$ candidate composite images for each $(\hat{I}, M)$ data pair. To ensure the quality of the generated images, we separately train a Harmony-VAE suitable for inverse harmonization model using the iHarmony4 dataset. Subsequently, we use the harmony classifier to classify the candidate set, selecting the image with the highest classification probability as the final composite sample.

## 5 EXPERIMENT

### 5.1 Datasets and Metrics

*5.1.1 iHarmony4 Dataset.* We conduct experiments on the benchmark iHarmony4 [10] dataset, which consists of four sub-datasets: HCOCO, HFlickr, HAdobe5K, and Hday2night, with a total of 65,742 (7,404 for testing) pairs of composite and real images in the training (testing) sets. Following previous work [10, 41], we merge the training set of the four sub-datasets into one entire training set and evaluate each sub-dataset individually.

*5.1.2 Human Harmony Dataset.* For the Human Harmony dataset, we filter the imaterial-fashion-2020-fgvc7 [12] dataset to remove images containing only products. After cleaning the dataset size is 29,106. We set all valid segmentation parts (except background) to 1 to construct foreground mask images. When building the Human Harmony dataset, we set $K = 10$ and use the harmony classifier to select the image with the highest classification probability. We divide the Human Harmony dataset according to the same training/testing set ratio as the iHarmony4 dataset, resulting in a training set of size 26,157 and a testing set of size 2,946.

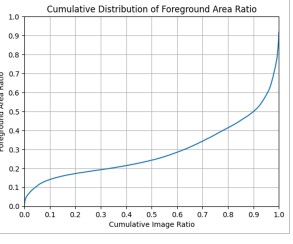 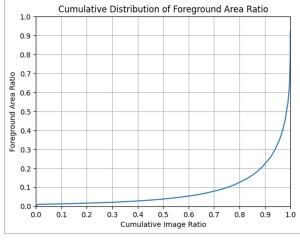

(a) Human Harmony      (b) iHarmony4

Figure 2: Cumulative distribution curves about foreground area ratios. We analyze and plot both iHarmony4 and Human Harmony dataset.

We analyze the distribution of foreground ratios and plot cumulative distribution curves. We also simultaneously plot the curve of iHarmony4. From Figure 2, it can be observed that approximately 70% of the images in the Human Harmony dataset have foreground ratios above 0.2, whereas in iHarmony4 this is less than 15%.

*5.1.3 Evaluation Metrics.* In line with previous works [16], our DiffHarmony++ and the baselines are evaluated using Peak Signal-to-Noise Ratio (PSNR), Mean Squared Error (MSE), and foreground MSE (fMSE) calculated across RGB channels. fMSE is a specific evaluation metric that solely measures the MSE within the foreground region, gauging the success of foreground harmonization.

### 5.2 Implementation Details

During the training of Harmony-VAE, we use all iHarmony4 training data. We set lr = $1e − 4$, with warmup = 0.02, and then keep it constant. The training lasts 10 epochs. We use the AdamW optimizer, with weight_decay=0, $\beta_1 = 0.9$, and $\beta_2 = 0.999$. We save model weights using exponential moving average (EMA), with max_decay = 0.999. Images are resized to 256px during training. As for the diffusion model, we adopt the pre-trained DiffHarmony [52] directly. Following [52], we use Euler ancestral discrete scheduler [21] to generate the samples in only 5 steps during inference.

The implementation details of inverse harmonization model are mostly consistent with DiffHarmony++, except that we use $(\hat{I}, M)$

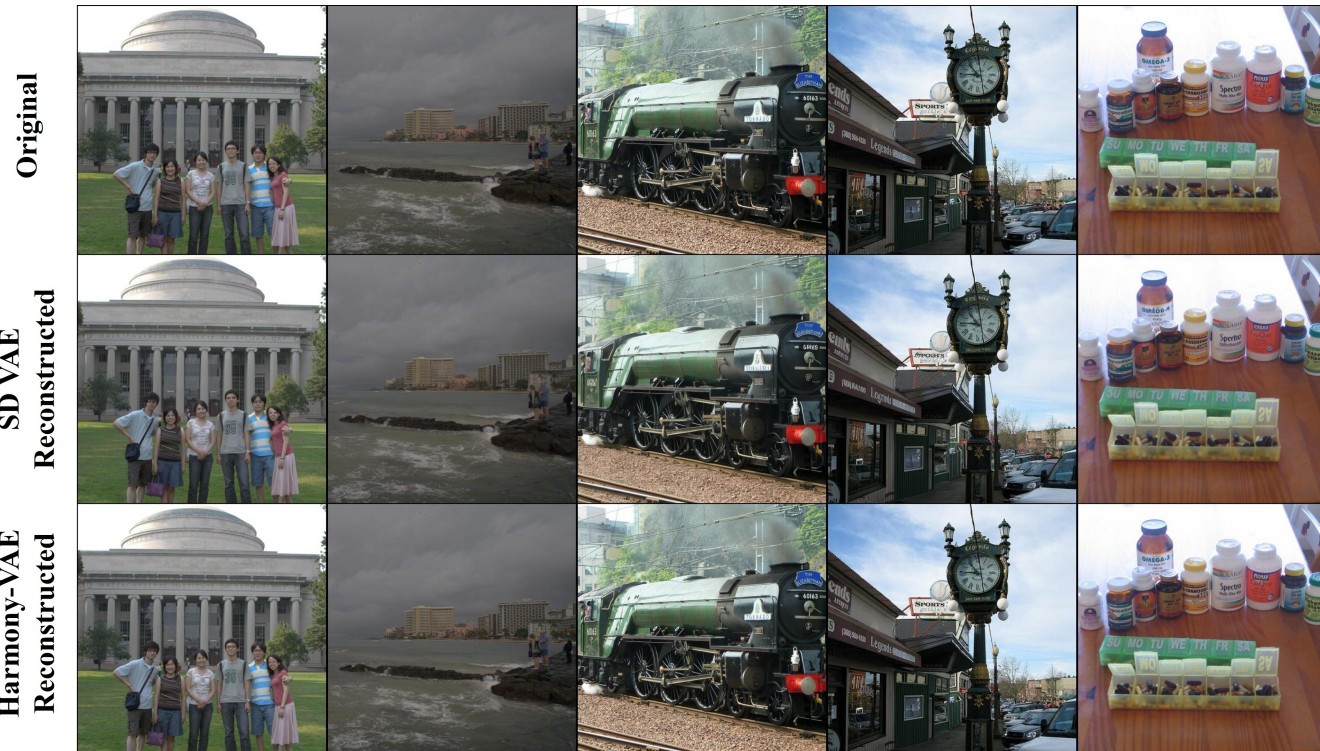

**Figure 3: Qualitative results of Harmony-VAE.**

as conditions and composite image *I* as the denoising target during training. The training and inference of corresponding Harmony-VAE is also modified accordingly.

For the harmony classifier, we utilize pre-trained ResNet50 [18] model, taking the final 2048-dimensional features after pooling for linear probing. We train a binary classification head using all iHarmony4 training data.

### 5.3 Performance Comparison

*5.3.1 Results on iHarmony4.* On the iHarmony4 dataset, we compare our approach with the following image harmonization methods: DoveNet[10] , BargainNet[8] , RainNet[30] , iS²AM[41] , D-HT[14] , SCS-Co[16] , HDNet[5] , Li *et al.*[28] , GIFT[35], DiffHarmony[52], etc. In Table 1, we report the average results for the four sub-test sets and the entire test set, which are either replicated from the original papers or reproduced using publicly available models. For the overall results on the entire test set, our method significantly outperforms previous SOTA methods. Our method achieves the best results on almost all sub-sets , except for the MSE on the HAdobe5k and Hday2night subsets.

Our proposed model outperforms DiffHarmony because we utilize Harmony-VAE instead of the VAE from Stable Diffusion. In the process of image decoding, using only the regular VAE from Stable Diffusion can result in severe distortion. In contrast, our proposed Harmony-VAE can preserve more details. Our qualitative results are provided in Figure 3. It can be observed that Harmony-VAE

successfully repairs severely damaged facial features, architectural patterns, small text, and other contents during the decoding process.

|  | base | augmented |
|---|---|---|
| Hday2night | PSNR: 39.06 | **PSNR: 42.26** |
|  | MSE: 25.70 | **MSE: 17.49** |
|  | fMSE: 439.11 | **fMSE: 223.67** |
| HFlickr | PSNR: 34.52 | **PSNR: 36.57** |
|  | MSE: 61.15 | **MSE: 41.18** |
|  | fMSE: 413.08 | **fMSE: 252.86** |

**Table 2: Qualitative results of data augmentation experiments. Data generated by inverse harmonization model can greatly boost model performace on small datasets.**

*5.3.2 Effectiveness of Data Augmentation.* We conduct data augmentation experiments with inverse harmonization model on two subsets of iHarmony4, Hday2night and HFlickr, as they have relatively fewer training data, making them more likely to benefit from data augmentation. HFlickr has 4,836 real images and 8,280 composite images, while Hday2night has only 109 real images and 447 composite images.

Our experimental results are shown in Table 2. "base" represents the results obtained by training only on the original dataset, while "augmented" represents the results obtained by training on the augmented dataset. For both datasets, we set the blending ratio $\gamma = 1$. All experiments use the same hyperparameter setting

| Dataset | Model | $0\% \sim 5\%$ | $5\% \sim 15\%$ | $15\% \sim 100\%$ |
|---------|-------|----------------|-----------------|-------------------|
| iHarmony4 | $N_{samples}$ | 3835 | 1951 | 1618 |
| | HDNet$_{512}$ | **PSNR: 45.64**
**MSE: 3.16**
**fMSE: 143.93** | PSNR: 39.97
MSE: 11.33
fMSE: 129.87 | PSNR: 34.59
MSE: 47.19
fMSE: 152.01 |
| | DiffHarmony++ | PSNR: 45.20
MSE: 3.67
fMSE: 173.40 | **PSNR: 40.17**
**MSE: 11.31**
**fMSE: 128.27** | **PSNR: 34.97**
**MSE: 41.94**
**fMSE: 136.67** |
| Human Harmony | $N_{samples}$ | 26 | 323 | 2597 |
| | HDNet$_{512}$ | PSNR: 41.05
MSE: 7.66
fMSE: 210.96 | PSNR: 37.16
MSE: 18.48
fMSE: 160.73 | PSNR: 32.60
MSE: 65.71
fMSE: 182.04 |
| | DiffHarmony++ | **PSNR: 42.48**
**MSE: 5.60**
**fMSE: 176.26** | **PSNR: 39.52**
**MSE: 11.45**
**fMSE: 101.86** | **PSNR: 34.31**
**MSE: 60.17**
**fMSE: 154.02** |

Table 3: Comparison between HDNet trained with high-resolution images and DiffHarmony++ on both iHarmony4 and Human Harmony dataset. Number of samples of every subset with different foreground proportions are denoted as $N_{samples}$.

and train with the same total amount of data. We train enough steps to ensure convergence and then evaluate on the original test set. We perform inference only on 512px images and then scale the results to 256px for evaluation, as achieving optimal inference performance is not necessary for observing the effectiveness of data augmentation. The results show that data augmentation with inverse harmonization model significantly improves the model performance on both datasets, thus validating its ability to generate high-quality composite images.

*5.3.3 Advanced Analysis.* In Table 3, we present a detailed comparison of the performance of DiffHarmony++ and HDNet on both the iHarmony4 dataset and the Human Harmony dataset. To ensure fairness, we stipulate that models directly utilize composite images as input during both training and inference. (Note: HDNet in its original implementation crops the background portion of real images as input background content, significantly improving its performance and resulting in unfair comparisons with other approaches.) Following HDNet[5], we divide data into three ranges based on the ratio of the foreground region area and the entire image: $0\% \sim 5\%$, $5\% \sim 15\%$, and $15\% \sim 100\%$. We calculate metrics for each range respectively. We train HDNet with 512px images, denoting it as HDNet$_{512}$. During testing, we utilize 1024px images as input and subsequently resize harmonized images to 256px for evaluation, maintaining consistent experimental settings with our approach.

Our results on the iHarmony4 dataset indicate that when test samples have small foreground proportions ($0\% \sim 5\%$), HDNet$_{512}$ outperforms DiffHarmony++. However, as the foreground proportion increases, DiffHarmony++ demonstrates increasingly superior performance. On the Human Harmony dataset, particularly on larger foreground proportions ($15\% \sim 100\%$), DiffHarmony++ consistently outperforms HDNet$_{512}$. Regarding its performance on samples with smaller foreground proportions, we speculate that the limited number of data samples leads to a significant variance in statistical metrics. Due to constraints in time and computational

resources, further investigation into the underlying reasons for this phenomenon is deferred for future research.

Qualitative results are shown in Figure 4. Our approach often generates visually appealing outcomes that closely resemble the authentic real images.

## 5.4 Ablation Study

We conduct ablation study to validate the effectiveness of the components in our proposed Harmony-VAE.

| $R_{inf}$ | Harmony-VAE | PSNR | MSE | fMSE |
|-----------|-------------|------|-----|------|
| 512px | ✘ | 38.12 | 25.09 | 292.16 |
| 512px | ✔ | 40.11 | 19.81 | 215.01 |
| 1024px | ✘ | 40.98 | 14.86 | 166.48 |
| 1024px | ✔ | 41.66 | 13.98 | 153.98 |
| 1024px | ✘ | 41.72 | 13.35 | 151.65 |
| 1024px | ✔ | 41.75 | 13.23 | 150.92 |

Table 4: Ablation study of using Harmony-VAE at multiple inference resolutions. For easier comparison we add additional results of DiffHarmony plus refinement module at bottom and mark the corresponding lines in gray color.

We do inference with and without Harmony-VAE at multiple resolutions. As shown in Table 4, adding the Harmony-VAE results in an improvement in the overall performance. The benefit of introducing Harmony-VAE is more prominent when DiffHarmony uses lower image resolutions, as the Harmony-VAE and using higher resolution input both aim to address the issue of image distortion, and they complement each other. We also add additional results of DiffHarmony plus refinement module [52] at bottom and mark the corresponding lines in gray color. Comparing to using only Harmony-VAE, using only the refinement module and the cascaded use of Harmony-VAE and refinement module only bring negligible improvement. This suggests that employing Harmony-VAE alone to enhance DiffHarmony could achieves optimal performance, and

| Original | Composite | HDNet | DiffHarmony++ |
|----------|-----------|-------|---------------|

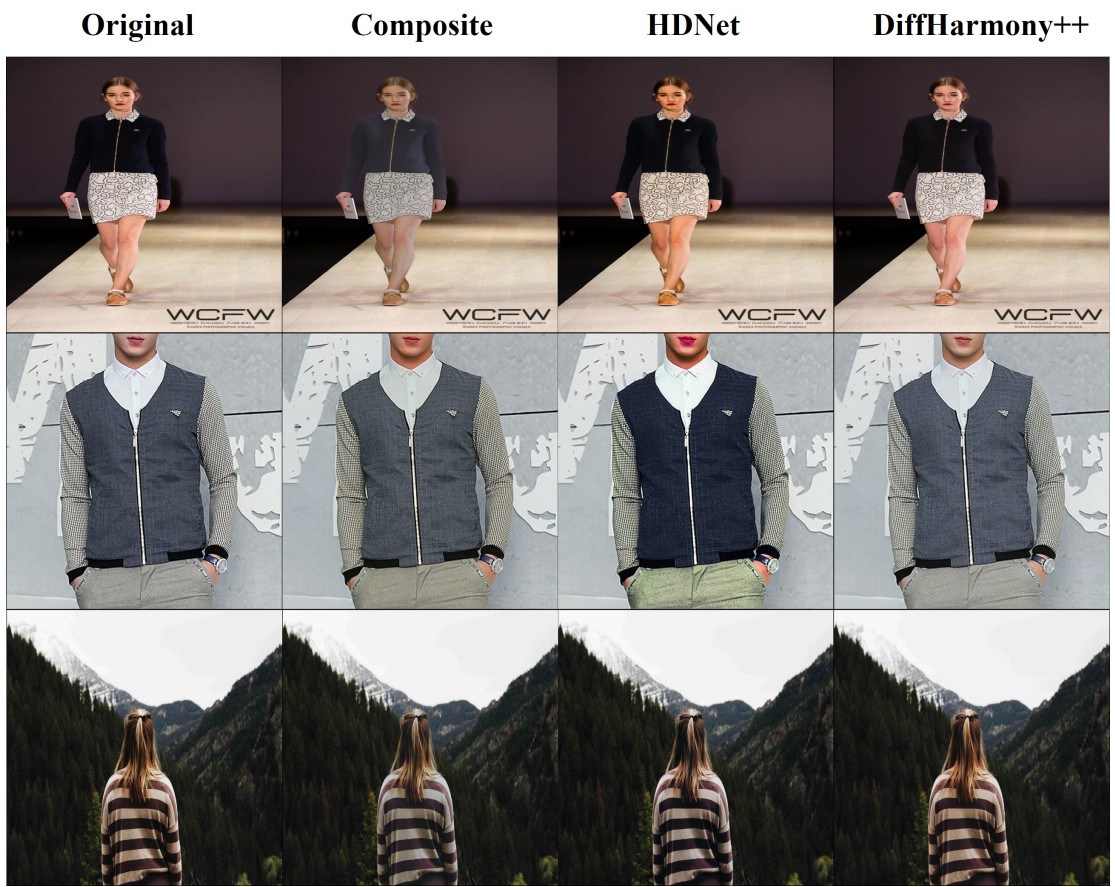

Figure 4: Qualitative results on Human Harmony Dataset.

furthermore Harmony-VAE can be trained in a more elegant and cost-effective manner. It is noteworthy that even training only on 256px images, the Harmony-VAE still brings significant improvement when inferring with 1024px images, demonstrating the generalization of our approach.

|  | zero init | random init |
|--|-----------|-------------|
| $\mathcal{E}_c$ | PSNR: 39.86
MSE: 20.45
fMSE: 221.05 | NaN |
| $\mathcal{E}_c + \mathcal{D}$ | **PSNR: 40.11**
**MSE: 19.81**
**fMSE: 215.01** | PSNR: 31.49
MSE: 114.81
fMSE: 1231.56 |

Table 5: Ablation study of our proposed zero init strategy and finetuning different parameters.

From Table 5, it can be observed that not using zero-initialized convolution severely damages the normal training of the model. The random initialization of convolution layers introduces nonnegligible perturbations to the original feature distribution when fused with the VAE decoder. From the fine-tuning parameter ablation

experiments, it can be seen that fine-tuning only the conditional encoder already provides a considerable improvement in reconstruction performance, while unfreezing the decoder can further increase the benefits.

## 6 CONCLUSION

In this paper, we have proposed the Harmony-VAE, aimed at leveraging the conditional information in image harmonization tasks to enhance the quality of image decoding of the VAE component in the latent diffusion model. Our proposed Harmony-VAE preserves finer details, effectively restoring severely damaged facial features, architectural patterns, small text during the decoding process. Furthermore, we have trained an inverse harmonization model which can synthesize new composite images based on real images and foreground masks. The substantial improvements observed on the Hday2night and HFlickr datasets attest to the efficacy of our model. Building upon this, we have subsequently constructed the Human Harmony Dataset, comprising samples with prominent foreground areas. Experimental results demonstrate the effectiveness of our inverse harmonization model of superiority of the LDM-based harmonization approach on samples featuring prominent foreground objects.

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
