# OpenReview forum: "DiffHarmony++: Enhancing Image Harmonization with Harmony-VAE and Inverse Harmonization Model"
_acmmm.org/ACMMM/2024/Conference — MM2024 Poster_

### Official Review · Reviewer_fxTh · 2024-05-23

**Rating:** 1
**Confidence:** 3

**Summary:**

This paper proposed an incremental VAE training strategy to improve the latent diffusion-based harmonization. The proposed Harmony-VAE could improve the high-frequency details for the original decoded results. This model can be also trained with inverse harmonization as a data augmentation for harmonization. Experiments show the effect of this VAE to preserve some foreground details.

**Strengths:**

1. The proposed method is straightforward and easy to follow.
2. The idea is simple but effective for latent diffusion-based manners.

**Limitations:**

1. The novelty is constrained. This work is more like an incremental work based on DiffHarmony to eliminate the bias of VAE decoder.
Moreover, the limitation of VAE decoder has been widely discussed, but this paper miss some important related works, such as [1][2].

[1] Zhu Z, Feng X, Chen D, et al. Designing a better asymmetric vqgan for stablediffusion[J]. arXiv preprint arXiv:2306.04632, 2023.

[2] Wang Y, Cao C, Fu Y. Towards Stable and Faithful Inpainting[J]. arXiv preprint arXiv:2312.04831, 2023.

2. The model designed in Fig1 suffers from train/inference biases. Because the training of Harmony-VAE uses GT inputs, while the inference uses denoised ones instead. The authors did not discuss how to address this issue.

3. The illustration of this paper is not well formulated. For example, it is really hard to identify which one is better in Fig.3 without the foreground mask. Moreover, there are no clear teasers to present and claim the contribution of this paper.

**Suitability:**

1

---

### Official Review · Reviewer_aCnh · 2024-05-24

**Rating:** 1
**Confidence:** 4

**Summary:**

This paper designs and trains a Variational Autoencoder (VAE) with an encoding-reconstruction task and then utilizes it in the Diffusion Model's VAE decoder to improve DiffHarmony's ability, namely DiffHarmony++. Moreover, this work is based on DiffHarmony++ to train an inverse harmonization model, which can produce composite images from real images and foreground masks.

**Strengths:**

This work automates the construction of composite images using an inverse harmonization model and validates its effectiveness on small datasets.

**Limitations:**

1. The proposed method's innovation is limited, as it primarily involves technical improvements to DiffHarmony, with its performance mainly derived from the effectiveness of the diffusion model. Compared to DiffHarmony, the performance improvement of DiffHarmony++ is not significant. Additionally, there is a discrepancy between the performance of DiffHarmony and the original paper, which is not explained.
2. The approach of automatically generating composite images is not novel, as the idea of using generative models to augment training data has been repeatedly discussed by research groups since the advent of GANs. Moreover, this paper only validates the effectiveness of generated data on a small dataset.

This paper lacks novelty and significant contributions, which are crucial for acceptance in ACM MM  papers.

**Suitability:**

2

---

### Official Review · Reviewer_aPGF · 2024-05-24

**Rating:** 5
**Confidence:** 4

**Summary:**

The paper introduces a sophisticated approach to improve the visual coherence of composite images. The authors propose Harmony-VAE to refine the decoding process of a latent diffusion model, addressing the distortion issues inherent in VAEs. Additionally, they present an inverse harmonization model for data augmentation, which creates synthetic composite images from real images and foreground masks. The paper also introduces a new Human Harmony dataset, enriching the multimedia community with resources for image harmonization tasks.

**Strengths:**

(1).Theoretical grounding: the paper is theoretically sound, with a clear explanation of how Harmony-VAE leverages additional input to improve image decoding. (2).Empirical validation: The empirical methodology is robust, with extensive experiments conducted on the iHarmony4 dataset and the newly created Human Harmony dataset, demonstrating the model's effectiveness. (3).Significance and novelty: the contributions of this paper are significant and novel. The Harmony-VAE presents a new way to tackle the distortion problem in VAEs, which is a critical issue in image harmonization tasks. The introduction of an inverse harmonization model for data augmentation is innovative.

**Limitations:**

(1).Depth of structural design analysis: The paper could benefit from a more in-depth analysis of the structural design choices. While the Harmony-VAE and inverse harmonization model are novel, there is a lack of detailed justification for the specific architectural decisions made. (2).Exploration of hyperparameter sensitivity: The paper does not delve into the sensitivity analysis of the hyperparameters used in the models. Understanding how changes in hyperparameters affect the outcomes is crucial for the robustness of the proposed methods.

**Suitability:**

3

---

### Official Review · Reviewer_jaVq · 2024-05-26

**Rating:** 4
**Confidence:** 3

**Summary:**

The paper proposes Harmony-VAE, leveraging the input of the harmonization task itself to enhance the quality of decoded images. The paper employs diffusion models to inverse image harmonization, i.e. generating synthetic composite images based on real images and foreground masks.
Extensive experimental results are given to demonstrate the effectiveness and performance of our proposed Harmony-VAE and inverse harmonization model.

**Strengths:**

The Harmony-VAE model is proposed to leverage the image harmonization task itself to enhance the quality of decoded images. This approach utilizes the precise pixel-level information contained in the composite image to complement the denoised latents, potentially improving the foreground appearance and color information.

An inverse harmonization diffusion model is propsed to to automatically construct new image harmonization datasets, which has long been a challenge in the field of image harmonization.

Experimental results are given to demonstrate the effectiveness of their proposed Harmony-VAE and inverse harmonization model, validating the proposed approach can achieve state-of-the-art performance on various image harmonization tasks.

A new dataset called Human Harmony dataset is delivered in the paper.

**Limitations:**

The paper lacks details about the exact network architecture, hyperparameters, and training procedures used for the Harmony-VAE and inverse harmonization models.  The hyperparameters in the implementation details are given emperically without any validation.

The evaluation methods used in the paper are not so convincing. We cannot see apparent differences of qualitative results in fig 3 and 4. The evaluation metrics used in paper are Peak Signalto-Noise Ratio (PSNR), Mean Squared Error (MSE), and foreground MSE (fMSE),  how does these metrics are related to image harmonization?

**Suitability:**

3

---

### Meta-Review · Area_Chair_r3Ub · 2024-06-26

**Recommendation:** Accept (Poster)
**Confidence:** 5

**Metareview:**

This paper receives a mixture of reviews. It is important to tackle image distortion when using diffusion model for image harmonization, which is addressed in this paper. This paper also makes the first attempt at using diffusion model for data augmentation in the field of image harmonization. Although similar ideas have been explored in other fields like image inpainting, the contribution to the field of image harmonization is non-negligible. The rebuttal has addressed the concern of train/inference biases. The visual results in the supplementary can justify the higher quality of the results generated by the proposed method. The authors are suggested to cite and discuss the two related papers mentioned by Reviewer fxTh.